# Transarterial Treatment of Lung Cancer

**DOI:** 10.3390/life12071078

**Published:** 2022-07-19

**Authors:** Atsushi Hori, Ikuo Dejima, Shinichi Hori, Shuto Oka, Tatsuya Nakamura, Shota Ueda

**Affiliations:** 1Institute for Image Guided Therapy, Izumisano 598-0047, Japan; horiat@igtc.jp (A.H.); okas@igtc.jp (S.O.); nakamurata@igtc.jp (T.N.); 2Department of Radiology, Wakayama Medical University, Wakayama 641-8509, Japan; ent2rea810n10v5@gmail.com (I.D.); sht.1210.tenten@gmail.com (S.U.)

**Keywords:** lung cancer, bronchial artery, antineoplastic agents, embolic material, microcatheter

## Abstract

Purpose: The treatment efficacy of the transarterial approach to lung cancer is evaluated. Materials and Methods: A total of 98 patients with advanced lung cancer or recurrent lung cancer after the standard therapies were enrolled retrospectively. The bronchial arteries and mediastinal branches from the subclavian artery were selected by a microcatheter. Immediately after the selective arterial infusion of anti-neoplastic agents, embolization with a spherical embolic material was carried out. Local tumor effects and overall survival were evaluated. Result: The mean reduction rate was 17.9%, with 24.2% for partial remission and with 2.1% for progression disease. The rate of stable disease was 72.6%. The response rate was 25.3%, and the disease control rate was 97.9%. The median survival time (MST) was 11.4 months, the 1-year survival rate was 45.2%, and the 2-year survival rate was 35.6%. Although it is insignificant, the MST for 51 adenocarcinomas was higher than that of 29 squamous cell carcinomas (18.6 months and 9.4 months, respectively). The local extension of tumors related to a better prognosis, though it was not significant. Lymph node metastases and distant metastases were poor prognostic factors. No major complications nor treatment-related mortalities were found in this study. Conclusion: The transarterial treatment for lung cancer should be considered as a treatment option when the other treatments were not indicated both in initial cases and in recurrent cases.

## 1. Introduction

With recent advances in cancer treatment, the outcomes of various malignant diseases have markedly improved. Although significant progress has also been achieved in the treatment of lung cancer in recent years due to advances in anti-cancer drugs, molecularly targeted agents, and immune checkpoint inhibitors, treatment options are often limited when the disease is diagnosed in highly advanced stages or recurrence occurs after standard treatments [1]. The number of lung cancer patients still continues to increase worldwide compared with other malignant diseases. Lung cancer treatment via the bronchial artery has a long history [2,3,4], but has long been neglected due to technical difficulties and advances in other lung cancer treatments [5]. However, recent advances in diagnostic imaging, catheter technology, and arterial embolization materials have led to significant advances in transarterial treatment, which can now be utilized for treatment via bronchial arteries [6].

This technique is less invasive for patients, and we hypothesize this technique can improve symptoms and prolong prognosis, which might make a significant contribution to the treatment of lung cancer. Herein, our institution has been involved in this treatment for a long period, and we report here 5 years of experience to evaluate the long-term clinical outcomes of transarterial treatment for advanced lung cancer to verify whether transarterial treatment contributes to improving treatment outcomes in lung cancer.

## 2. Materials and Methods

### 2.1. Patients

From October 2016 to December 2021, 98 patients who had advanced lung cancer or recurrence after standard treatments were investigated retrospectively. They were treated with transarterial methods and were followed up by a CT at least one month after treatment. The mean age of patients was 66.5 years (46–87); 65 male cases, and 33 female cases. Previous treatments in other hospitals included systemic chemotherapy (65 patients), radiotherapy (22 patients), and surgery (13 patients). The initial treatment was performed on seventeen patients due to advanced stages for chemotherapy or radiotherapy. Histological examination showed 83 cases of non-small cell lung cancer (52 cases of adenocarcinoma and 31 cases of squamous cell carcinoma), 14 cases of small cell lung cancer, and 1 case of large cell lung cancer (Table 1).

This retrospective study was approved by our IRB.

### 2.2. Treatments

A contrast-enhanced dynamic CT (64-row MDCT) scan was performed as diagnostic imaging before treatment, and a 3D reconstruction (TeraRecon, Durham, NC, USA) to identify tumor feeders from the subclavian, intercostal, and inferior phrenic arteries, as well as the bronchial arteries directly from the aorta. In all patients, a 4Fr guiding catheter (Medikit, Tokyo, Japan) was inserted through the inguinal artery and a microcatheter (Estream IGT, Toray, Tokyo, Japan) was selectively advanced into the tumor-feeding arteries under the guidance of the vascular 3D images. After selective digital subtraction (DSA) imaging of the individual artery, a CT was performed while injecting contrast through the microcatheter to confirm blood flow to the tumor and lymph nodes. The drugs used were anti-neoplastic agents (doxorubicin; 20–30 mg, fluorouracil; 250–500 mg, cisplatin; 10–40 mg, gemcitabine; 100–200 mg, irinotecan; 20–40 m, docetaxel; 10–20 mg) and angiogenesis inhibitors (BV; bevacizumab, Chugai, Tokyo, Japan) with reference to patient’s prior treatment history. Several drugs among these were selected and the dose of each drug was about 20–40% of systemic chemotherapy. We selected as many tumor-supplying arteries as possible and distributed the drugs according to the proportion of tumor blood supply in each artery. Arterial embolization was performed immediately after chemotherapeutic drug infusion. The embolization procedure was terminated when the blood flow became slow. The arterial embolization material used was a spherical drug-eluting embolic material, HepaSphere (50–100 micron) (Merit Medical, South Jordan, UT, USA). Treatment efficacy was evaluated by a CT 1-month post-treatment.

If the treatment was deemed effective, treatment with the same drugs was repeated, otherwise, the drug was changed for transarterial treatment. Treatment was continued monthly for up to three sessions. Patients were then followed up and treated in the same manner when tumor progression was observed. No concomitant systemic chemotherapy was administered during the transarterial treatment.

### 2.3. Evaluation Methods

Tumor response was determined after 1 month by RECIST 1.1. The survival time was defined as the time from the initial date of the transarterial treatment to the latest follow-up date. The survival curves were estimated by the Kaplan–Meier method, and the difference in survival by age, gender, tumor histology, and TMN stages was tested by the log-rank test in which a *p*-value of less than 0.05 was considered significant.

## 3. Results

A total of 95 patients with measurable lesions were evaluated for tumor reduction rate (Figure 1). The mean reduction rate was 17.9%, with 24.2% of partial remission (23 patients showed a higher than 30% reduction) and with 2.1% of progression disease (2 patients showed a higher than 30% increase). The rate of stable disease was 72.6% (69/95). The response rate was 25.3% (CR + PR), and the disease control rate was 97.9% (CR + PR + SD). 

Survival curves for the 98 patients who could be followed for at least 1 month are shown in Figure 2. The median survival time (MST) was 11.4 months, the 1-year survival rate was 45.2%, and the 2-year survival rate was 35.6%. Survival in females (Figure 3) was significantly higher (*p* = 0.193). Ages over 50-year-old did not show significant differences. Previous therapies did not affect the prognosis (Figure 4). Although it is insignificant (*p* = 0.3520), the MST for 52 adenocarcinomas was higher than that of 31 squamous cell carcinomas (18.6 months and 9.4 months, respectively) (Figure 5). Small-cell lung cancer had a poor prognosis in 14 patients with MST of 6.3 months. Local extension of tumors related to better prognosis (Figure 6), though it was not significant (*p* = 0.1657). Lymph node metastases and distant metastases were poor prognostic factors (Figure 7 and Figure 8). Advance d clinical stages were also poor prognostic factors (Figure 9). No major complications nor treatment-related mortality were found in this study.

## 4. Discussion

Transarterial treatment for lung cancer has long been reported to improve local effects, reduce drug dosage, and decrease side effects has long been attempted [1,4,7,8]. However, it did not become a standard treatment method due to its complexity and uncertain drug distribution to the target lesions [5]. Recent advances in MDCT technology and 3D imaging together with the advent of spherical embolic material and microcatheter have made the transarterial treatment an alternative treatment for advanced lung cancer [1,6,9,10], and serious side effects are less likely to occur. In our present study, we demonstrated remarkable technical success without major complications or treatment-related mortality.

Identifying the blood supply to lung and mediastinum cancers is the key to success. Several reports have shown that the bronchial arteries and other systemic arteries, but not the pulmonary artery is the main tumor-feeding artery for lung tumors [1,3,4]. As the arterial pressure is higher than the pulmonary arterial pressure, the dual blood supply from the pulmonary and bronchial arteries is unlikely. The bronchial artery normally arises directly from the descending aorta or the aortic arch, especially on the right side, the right bronchial artery often forms a common trunk with the upper intercostal artery. Some branches may be arising from the proximal origin of the brachiocephalic or the left subclavian artery. Tumors and lymph node metastases in the mediastinum are often supplied by branches from the thyrocervical artery or from the internal thoracic artery [5,11]. There may be anastomotic branches from the inferior phrenic artery that penetrates the diaphragm. If the tumor invades the chest wall, blood flow is dominantly from the intercostal arteries [8]. 

Careful attention should be paid to the treatment procedures involving the right bronchial artery, which often arises from the right intercostal artery, and the branches from the subclavian artery. Overflow of embolic materials to the intercostal artery should be avoided. Analysis of angiographic images in combination with angio-CT is indispensable [6]. In our experience, there were no cases of neurological symptoms or complications even when the drugs were administrated into the intercostal artery. Although bronchial mucosal damage due to high doses of CDDP infusion has been reported [7], the maximum dose in this report was limited to 20 mg and no clinical symptoms were experienced. However, the optimal dosage and type of drug need to be determined in more cases. Vascular shunts from the bronchial artery to the pulmonary artery or pulmonary vein may be observed in the periphery of the lungs [12]. If a shunt to the pulmonary vein is suspected, embolization should not be performed to avoid systemic embolization [6].

To obtain a good local effect, it is necessary to allocate the amount of drugs to be injected according to the blood supply ratio [6,13]. In the present study, most patients were systemic chemotherapy-refractory, and the chemotherapeutic drugs selected were based on patients’ prior treatment. There was no local irritation during the infusion of drugs except for gemcitabine which may cause pain during infusion if the concentration is higher than 5 mg/1 mL. As for the dosage, since it was less than 1/4 of the systemic chemotherapy dosage, the side effects of the anticancer drug were minimal, and only the usual antiemetic was administered. This treatment has been reported previously [1,5,8,9,12,14,15] to cause no serious complications, confirming the safety of this treatment. The optimal drug selection and dosage should be studied in the future, as it depends on various factors such as tumor burden, tumor morphology, and response to previous drugs.

The survival rate in this study was lower than our previous results [6], this may be due to 88.8% lung cancer Stage III and above, and the longer observation period in this study. However, the MST was almost 1 year despite the highly advanced lung cancer, mainly recurrent cases. Although comparisons are extremely difficult due to the different backgrounds of previous reports, they all lasted about 1 year [1,5,8,16] and do not differ significantly from previous results. In our subgroup prognostic study, the less local extension, and absence of lymph node metastases or distant metastases extension, were better prognostic factors that align with the prognostic statistics for lung cancer overall [17]. The transarterial treatment for lung cancer might be considered a treatment option when the other treatments were not indicated both in initial cases and in recurrent cases. The prognosis seemed better for adenocarcinoma in this study. This may reflect the fact that adenocarcinoma generally has a better prognosis than squamous cell carcinoma [17].

Since the present study was not a comparative single-center study, it is difficult to evaluate the results objectively. Based on our results, it is desirable to study more cases at multiple centers in the future. The important objective of this treatment is to obtain improvement in patients’ symptoms, but because this was a retrospective study, we were unable to accurately assess changes in symptoms. In the future, a prospective study should be conducted to confirm the initial efficacy of the treatment.

## 5. Case Presentation

Case 1: A 67-year-old man consulted our clinic with a local recurrence of lung cancer in the left pulmonary hilum (Figure 10). He had got systemic chemotherapy and radiotherapy. Symptoms at consultation were severe cough and respiratory distress. The first treatment through the bronchial arteries was carried out by infusing 5-FU; 250 mg, CDDP; 20 mg, DOC; 20 mg, BV; 200 mg with embolization by 3.0 mg of HepaSphere (50–100 micron). In total, 8 sessions of treatment have been repeated. The patient at present (April 2022) has no clinical symptoms except for a slight cough with no clinical problems in the activity of daily life (ADL).

Case 2: A 75-year-old woman consulted our clinic complaining of severe cough (Figure 11). A Tumor located in the proximal part of the right upper lobe caused atelectasis of the right upper lobe. She had not undergone systemic chemotherapy due to her age and concomitant bronchiectasis. The first treatment through the bronchial arteries was carried out by infusing docetaxel; 20 mg, cisplatin; 20 mg, bevacizumab; 100 mg with embolization by 1.0 mg of fluorouracil loaded HepaSphere (50–100 micron). In total, 6 sessions of treatment have been repeated to maintain better QOL. The patient died from obstructive pneumonia 29 months after the initial treatments.

Complete disappearance of the target lesion was found and intrapulmonary metastases in the right lung were newly detected.

## Figures and Tables

**Figure 1 life-12-01078-f001:**
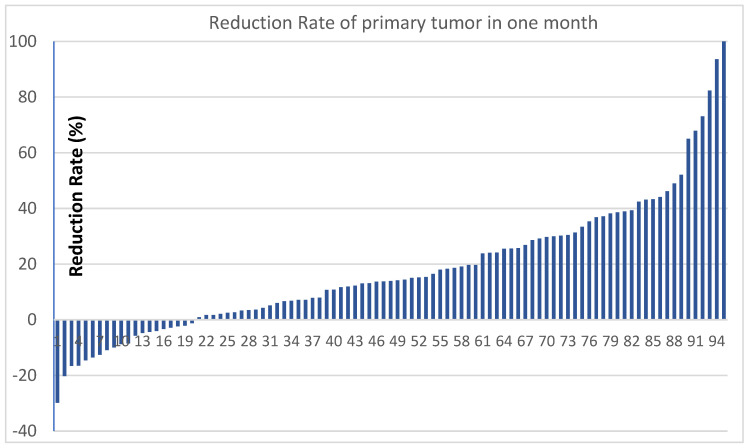
Tumor reduction rate in one month.

**Figure 2 life-12-01078-f002:**
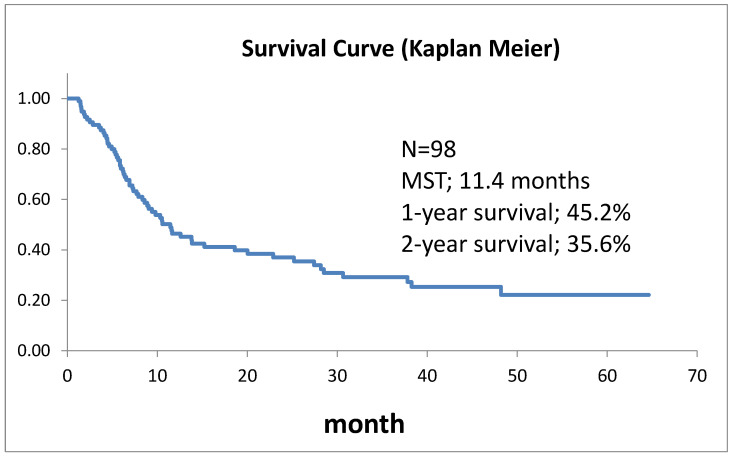
Survival of 98 patients with lung cancer treated by trans-arterial treatment.

**Figure 3 life-12-01078-f003:**
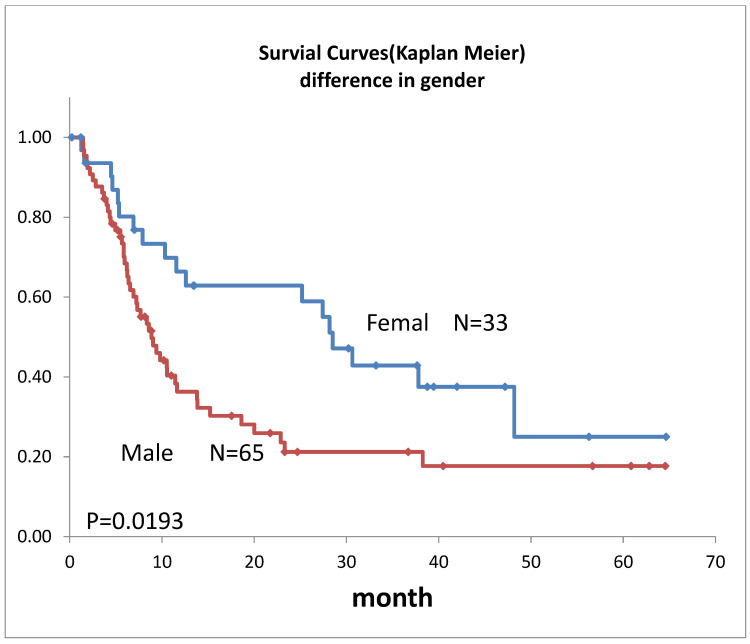
Survival of lung cancer patients with the difference in gender.

**Figure 4 life-12-01078-f004:**
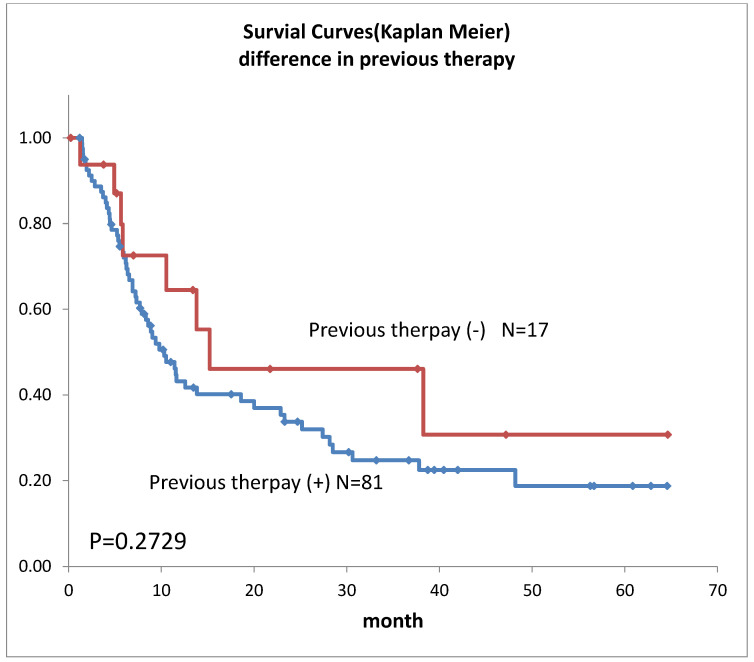
Survivals of lung cancer according to previous therapies.

**Figure 5 life-12-01078-f005:**
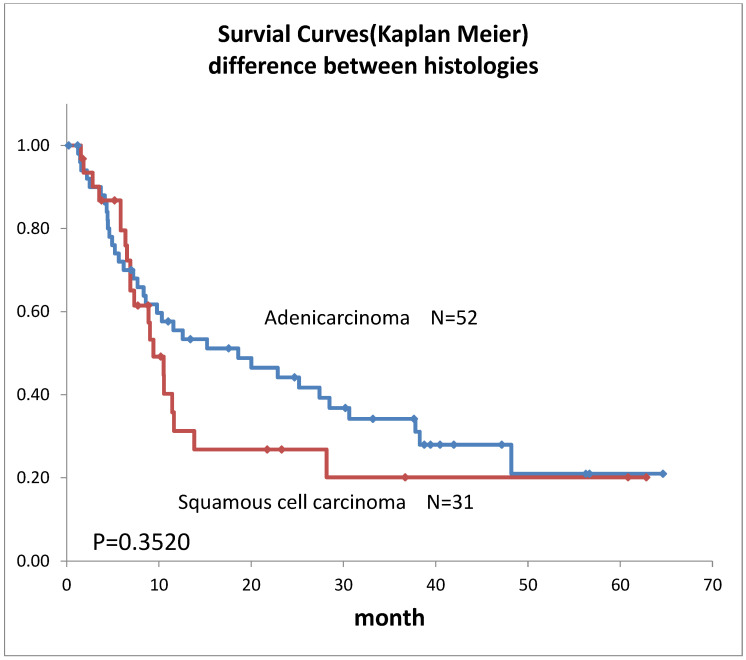
Survivals of lung cancer with the difference in histology.

**Figure 6 life-12-01078-f006:**
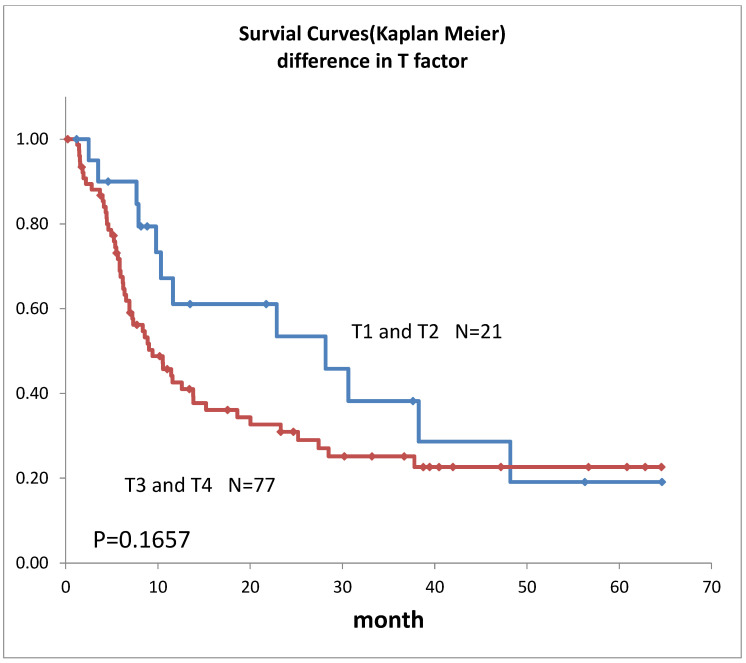
Survivals of lung cancer with the difference in T factors.

**Figure 7 life-12-01078-f007:**
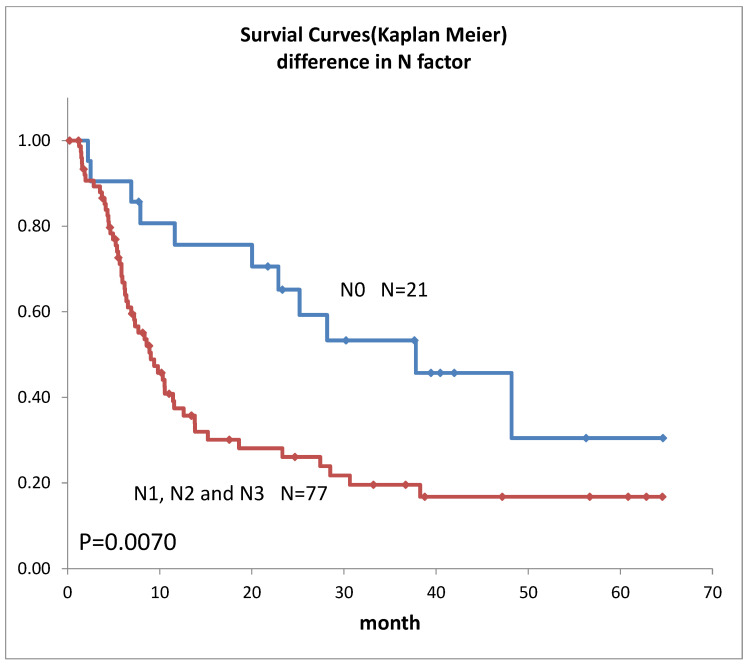
Survivals of lung cancer with the difference in N factors.

**Figure 8 life-12-01078-f008:**
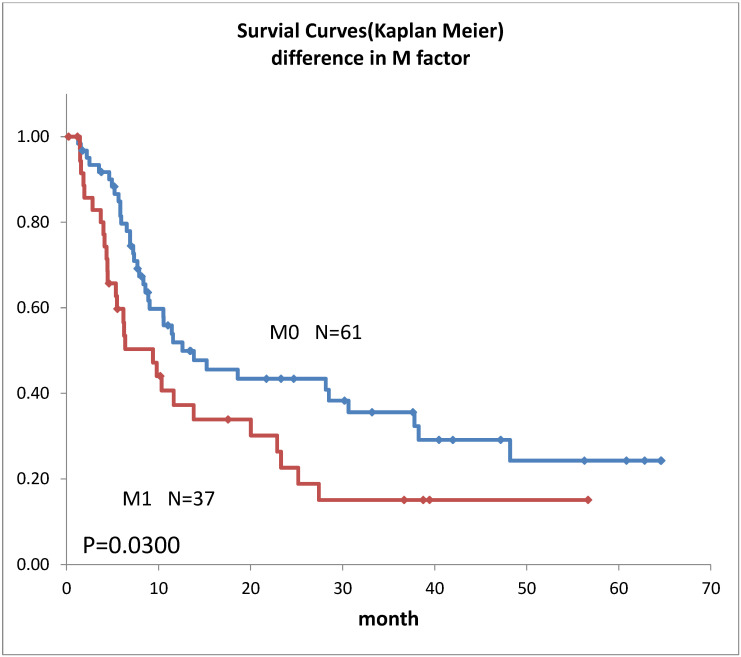
Survivals of lung cancer with the difference in M factors.

**Figure 9 life-12-01078-f009:**
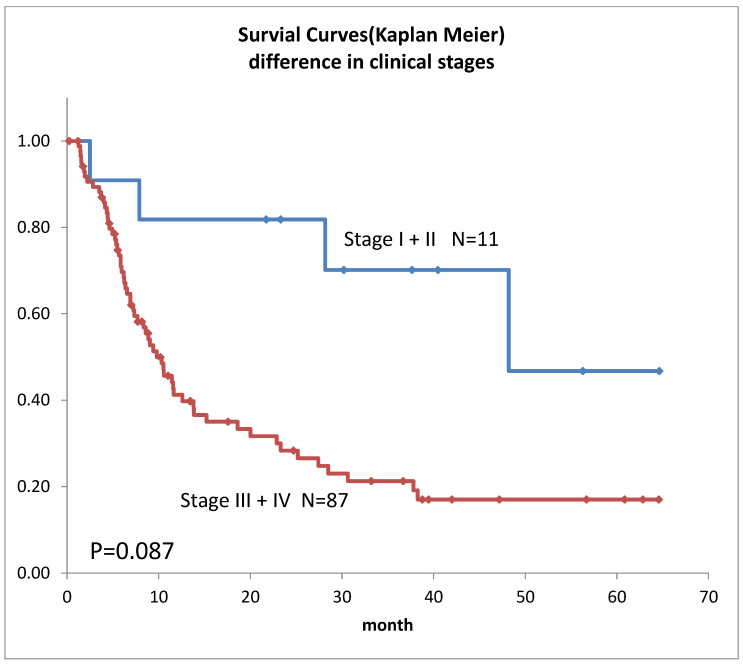
Survivals of lung cancer according to the clinical stages.

**Figure 10 life-12-01078-f010:**
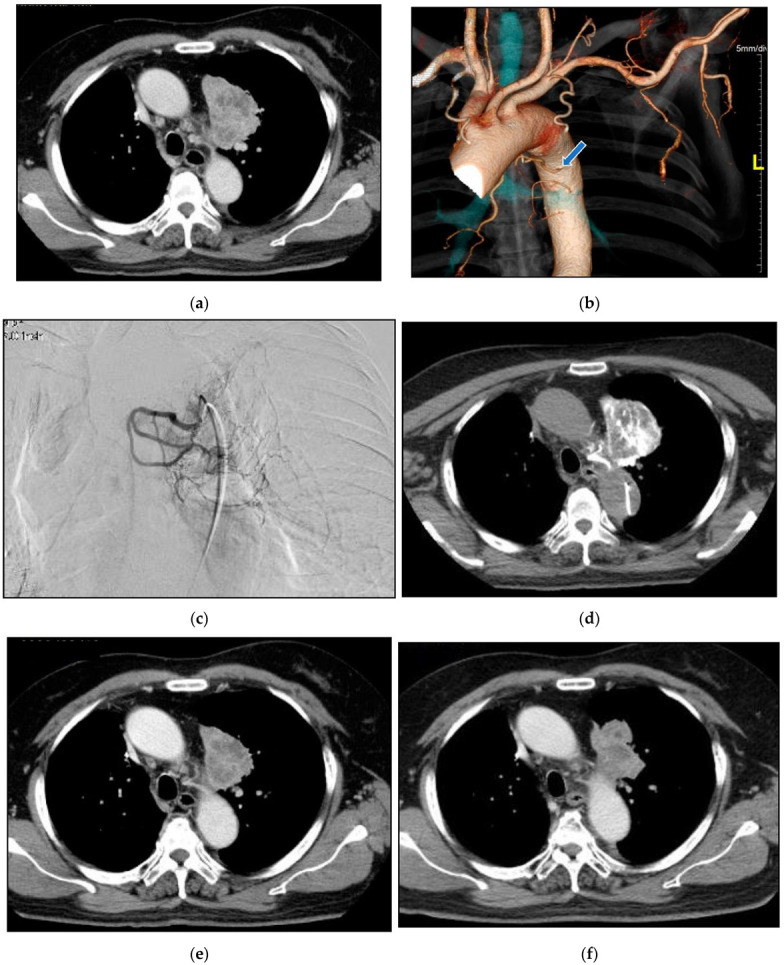
Recurrent adenocarcinoma in the lung. 67-years-old man. (**a**): Contrast-enhanced CT A tumor in the left upper lobe invading mediastinum with mediastinal lymph node metastases. The longitudinal diameter was 67 mm. (**b**): A volume rendering 3D image of the aorta and its’ branches. The main left bronchial artery (arrow) was clearly shown to arise from the anterior wall of the descending aorta. (**c**): Selective DSA (Digital Subtraction Angiography) of the left main bronchial artery. It was difficult to assess the tumor supply from the bronchial arteriography. (**d**): Angio-CT of the bronchial artery. The blood supply to the whole tumor was recognized by infusion of contrast into the left bronchial artery. (**e**): A contrast-enhanced in a month after the initial therapy. Tumor reduction was observed. The diameter was 50 mm. (**f**): A contrast-enhanced CT in one year after the initial therapy. The diameter was 51 mm. Six sessions of treatment were carried out in one year. (**g**) A contrast-enhanced CT in 20 months after the initial therapy. The diameter was 56 mm. Eight sessions of treatment were carried out in the clinical course.

**Figure 11 life-12-01078-f011:**
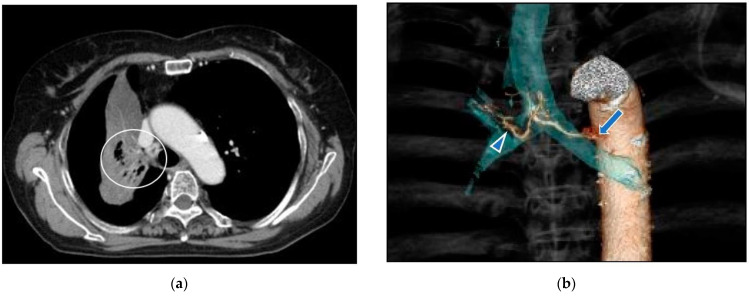
A 75-year-old female with right lung cancer. (**a**): A contrast-enhanced CT. The tumor (circle) around the right upper bronchus caused atelectasis of the right upper lobe. (**b**): A volume rendering 3D image demonstrates the common trunk (arrow) of the bronchial and intercostal artery. The right bronchial artery (arrowhead) distributes along the right upper bronchus. (**c**): Selective DSA of the right main bronchial artery. Abnormally dilated vessels were demonstrated around the right upper bronchus. (**d**): Angio-CT of the right bronchial artery. The tumor causing atelectasis of the right upper lobe was enhanced. (**e**): A contrast-enhanced CT in a month after the initial therapy. Shrinkage of the tumor was found and atelectasis was improved. (**f**): A contrast-enhanced CT in 20 months after the initial therapy. Six sessions of treatment were repeated in the same manner.

**Table 1 life-12-01078-t001:** Patients Characteristics.

Patients’ Characteristics	Number
Age;	66.5 (46–87)
Male/Female	65/33
Histology;	
Adenocarcinoma	52
Squamous cell carcinoma	31
Small cell carcinoma	14
Large cell carcinoma	1
Initial tumor treatment	17
Recurrent tumor treatment	81
Previous therapy;	81
Systemic chemotherapy	65
Radiotherapy	22
Surgery	13
None	16
T factors;	
T1	7
T2	14
T3	28
T4	49
N factors;	
N(−)	21
N(+)	77
M factors;	
M(−)	61
M(+)	37
Clinical Stages;	
Stage I	8
Stage II	3
Stage III	50
Stage IV	37

## Data Availability

Not applicable.

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
