# Peer review of "Transarterial Treatment of Lung Cancer"

_life, 2022, doi:10.3390/life12071078_

Round 1

Reviewer 1 Report

The study is quite relevant and interesting. However, due to the limited sample size, the differences in prior treatment management and histology type of disease, it makes hard to draw a strong conclusion.

Specific comments:

1.     Cited references need to be formatted according to journal specification.

2.     Predicting cases with progression disease is preferable, any characterizations to be rolled out?

3.     Stable disease rate was also high, any characterizations to be rolled out?

4.     98 patients is a small sample if to be characterized based on prior management, respectability and histological disease type. Hence, its hard to draw a conclusion.

5.     Recent advance in MDCT has promoted trans-arterial treatment, in what aspect?

6.     So the novelty of this work relies on the trans-pulmonary artery, right! What was the prognosis rate for the studies that used bronchial and other systemic arteries instead?  

Author Response

Reviewer 1

  1. Cited references need to be formatted according to journal specification

Corrected:  According to journal specification, the format of citations was corrected.

  1. Predicting cases with progression disease is preferable, any characterization to be rolled out?

Predicting factors with the progression disease could be histological type, sizes of target lesions, or response to drugs.  It seems difficult to mention the predicting factors in this small sample.

  1. Stable disease rate was also high, any characterization to be rolled out?

The rate of size reduction was evaluated one month after the initial treatment.  Many cases were treated by multiple sessions.  The rate of PR could be much higher if evaluated in three months. 

  1. 98 patients is a small sample if to be characterized based on prior management, respectability and histological disease type. Hence its hard to draw a conclusion.

Corrected;  the affirmative expression, in conclusion, was changed.

  1. Recent advance in MDCT has prompted trans-arterial treatment, in what aspect?

Due to the shorter scan time, the arterial phase of contrast enhancement can be taken efficiently. The thickness of each slice is less than 0.6mm.  The 3D image workstation offers excellent vascular images before trans-arterial treatment.  They are very helpful to prepare the best catheter or decide a treatment plan.

  1. So the novelty of this work relies on the trans-pulmonary artery, right! What was the prognosis rate for the studies that used bronchial and other systemic arteries instead?

We also believe that the novelty of this work relies upon the trans-bronchial approach.  We could not find previous data mentioning prognosis as to the pulmonary artery access.

Reviewer 2 Report

Very interesting, complete study, with excellent bibliographic references and very satisfactory iconography; perhaps the sample of patients is a bit heterogeneous by type of pathology, but all in all it would be difficult to enlist a sufficient number of patients by analogy of pathology; so the problem can still be overcome

Author Response

As indicated, we have to continue to evaluate the treatment efficacy with a larger sample.

It would be better to do a multi-center study to confirm the validity of this study.